# Driver's Social Relationship Based Clustering and Transmission in Vehicle Ad Hoc Networks (VANETs)

**Lin Li [1,2], Wenjian Wang [1,3],\* and Zhenhai Gao [2]**

[1] School of Computer and Information Technology, Shanxi University, Taiyuan 030006, China; lilynn1116@sxu.edu.cn
[2] State Key Laboratory of Automotive Simulation and Control, Jilin University Nanling Campus of Jilin University, Changchun 130012, China; gaozh@jlu.edu.cn
[3] Intelligence and Chinese Information Processing, Shanxi University, Taiyuan 030006, China
\* Correspondence: wjwang@sxu.edu.cn

**Abstract:** Clustering is a technique for dividing a network into different group of nodes and managing the transmission of data among the interacting nodes, to improve the effectiveness and safety of information transfer. Clustering have been well studied and applied in traditional mobile networks. However, vehicle networks have short connection time, frequently changing topology, and other unique properties that conventional clustering cannot transfer well. The vehicle nodes in Vehicle Ad-hoc Networks (VANETs) are most directly affected by the surrounding vehicle nodes and exchanged information with them. However, this will cause network congestion or even the spread of malicious messages. The inclusion of vehicle's (driver's) social relationships in vehicle communication clustering will increase the degree of trust between vehicle nodes, making communication more purposeful and accurate. This study proposed a new clustering for vehicle networks that is based on drivers' social relationship combined with the instantaneous position and speed of the vehicle node. Simulation results showed that this clustering method can improve the effectiveness of information transmission and increase the utilization of the application layer.

**Keywords:** clustering; vehicle network; drivers' social relationship

## 1. Introduction

Vehicle networks enable convenient information sharing and exchange among vehicles and road infrastructure. However, vehicle networks' particular features like high moving speed and frequent partition due to high mobility can invalidate or lose information. The safety and effectiveness of information transfer has become a research focus to facilitate better information sharing. Clustering technology divides network nodes into different clusters to improve the effectiveness of information transmission, and it has been well studied and used in traditional mobile networks [1], while a mobile network refers to a network with mobile nodes, including the network formed by the mobile nodes, such as people and mobile devices. Although a vehicle network is a special mobile network, the traditional network clustering approach is not applicable to it.

According to the short connection time and fast change speed unique features of vehicle networks, many scholars have proposed different clustering methods in a vehicle network environment, such as roughly static clustering based on road base station and dynamic clustering based on vehicle nodes. In static clustering, base station serves as the cluster head [2]. Nearby vehicles and the base station exchange information, and the base station passes this information to other vehicles. Although this clustering method is easy to distinguish, the distance between the base stations are large, so many issues may

affect information exchange. In addition, because vehicle network topology changes frequently, static clustering will delay the information transfer and reduce information accuracy. By contrast, dynamic clustering divides vehicles by certain rules [3], including rules of position, speed, and destination, and can customize the relationship of vehicle attributes. For example, bus and taxi [4] fleets cluster in a self-contained group. However, dynamic clustering may lack interaction (drivers' social relationship). We believe that there should be vehicle node relationship driven by human dynamics because many studies have confirmed that human dynamics is a significant factor in the social network of the people involved.

Clustering is an important research topic in MANETs (Mobile Ad-hoc NETworks) because it improves the system performance of large MANETs. Figure 1 shows an example of clusters in the traditional mobile network clustering [5], different cluster head selection algorithms are studied, and how to choose cluster head in a particular cluster is analyzed to extend network lifetime and improve the reliability of data transmission. Bednarczyk et al. [6] proposed a clustering method based on weight coefficients in MANETs, and the algorithm has a hierarchical structure that can maintain topology of MANET as stable as possible, thereby optimizing network performance and making efficient resource allocation for nodes. This makes the MANET environment topology more efficient and stable. Rajkumar et al. [7] suggested a clustering based on geographical factors in MANETs, which decided a baseline between the source cluster head and the destination node for route discovery. This method performed for the network with a high mobility of nodes.

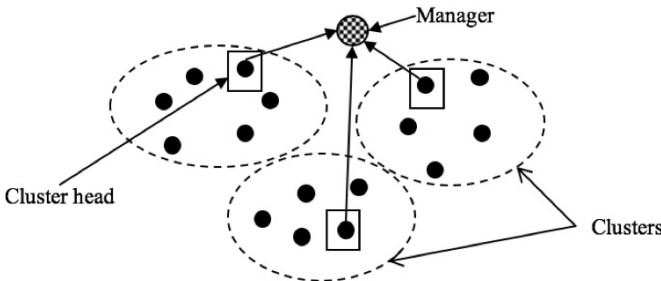

**Figure 1.** Clusters in traditional mobile network clustering.

VANET is a special class of MANET that differs from the latter in the number of parameters. Clustering and cluster-based routing protocols are the two important areas currently being studied, especially in VANET [8]. In [9–13], clustering in vehicle networks is studied. With the aim of efficient information propagation, Kanemaru et al. [9] proposed a vehicle clustering algorithm based on the similarity of geographic locations and trajectories. Washio et al. [10] put forward a vehicle clustering algorithm based on geographic location and trajectory to improve information validity in the communication process. Ucar et al. [11] designed the vehicular multi-hop algorithm for stable clustering (VMaSC), which combined the cluster with the vehicle moving speed. Arkian et al. [1] proposed a new clustering mix of static (road infrastructures) and dynamic (vehicles) nodes according to the parameters of different network layers. Li et al. [12] attempted to summarize the relevance of vehicle nodes in the VANETs and divided the vehicles into different clusters according to relevance. They established the correlation coefficient based on the position and velocity, which can distinguish vehicles according to road construction. Vodopivec [13] created a multi-agent support rule and a new clustering algorithm wherein the vehicle nodes in a cluster can send and receive messages on their own without relying on other nodes or the road infrastructure to recognize their located group; clusters are created with redundant connections between nodes to increase the communication reliability in case of topological changes.

Clustering for vehicle networks is not completely adapted from traditional mobile networks. In vehicle networks, finding or defining the cluster head in the clustering algorithm is difficult because each vehicle features the same properties. In addition, the existing vehicle networks clustering

has little regard to the impact of drivers' dynamic factor. Studies have shown that the trajectory of human behavior has great regularity in time and space, and people perform activities with large frequency in a narrow range of areas or fixed time interval [14]. Therefore, It is believed that vehicles driven by people follow the same law, and drivers' social group must be considered in clustering of VANETs. Yi et al. [15] using the individual drivers trajectories for learning in-depth driving behaviors, distinguish low-level and high-level driving behaviors to develop a personalized driver intention prediction system at unsignalized T intersections by seamlessly integrating clustering and classification.

To introduce drivers' dynamics to vehicle communication clustering, a social group-based vehicle clustering method is proposed. First, the drivers are divided into small groups to form a social network of driver groups. Then, the correlations of vehicles' position and velocity among nearby traveling vehicles are established. Finally, the relevant social relationship of vehicles and correlation between position and speed are combined. According to traffic rules after clustering, the communication among vehicles based on the proposed new clustering method and compare the benefits with other communication parameters (e.g., the benefits of the application layer, channel utilization, and throughput) is simulated.

The remainder of this paper is organized as follows. Section 2 presents the Materials and Methods. Section 3 shows the results of the proposed methods in detail, followed by the discussion in Section 4. Finally, Section 5 draws the conclusion and declares our next work.

## 2. Materials and Methods

Each vehicle node is defined a quintuple [A,N,G,M,C] as clustering management where: A represents the regional codes using the method of Quadkey [16]. The Quadkey reproduces the projection, coordinates systems, and addresses the scheme of the map. The Quadkey document represents the map at many levels of detail, and to cut each map into tiles for quick retrieval and display. In Quadkey, the length of a quadkey (the number of digits) equals the level of detail of the corresponding tile, and the quadkey of any tile starts with the quadkey of its parent tile (the containing tile at the previous level), as shown in Figure 2. Quadkey encoding is used for drivers' social location; N is the number of vehicles within their region, for example, $N_i$ = 1000 means there are 1000 vehicle nodes within the region of the *i*-th node. G (1,2,3,...G) is the number of the group that one node belongs to, and the number of a certain group is indicated by the minimum number of the node in this group. That is to say, $g_i = j$ indicates that the number of current group is *j*, and *j* is the node with the minimum number in this group $g_i$; M is the number of vehicle nodes of the current node that belongs to group G; and C is the cluster set of nodes in the current cluster. A flowchart as shown in Figure 3 depicts the steps of the research and the final target.

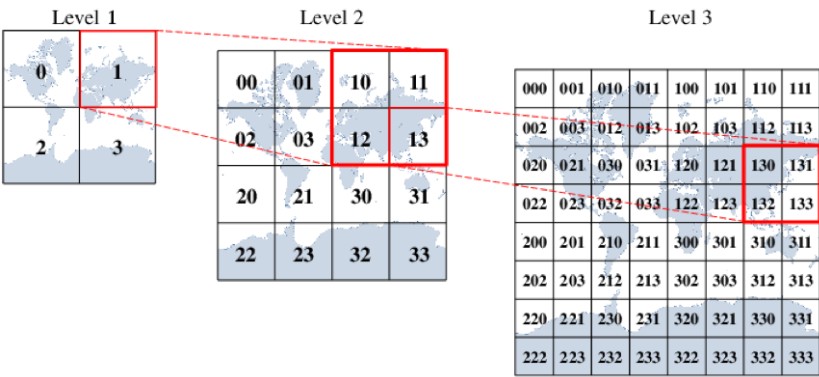

**Figure 2.** Hierarchical structure and inclusion relation of Quadkey [16].

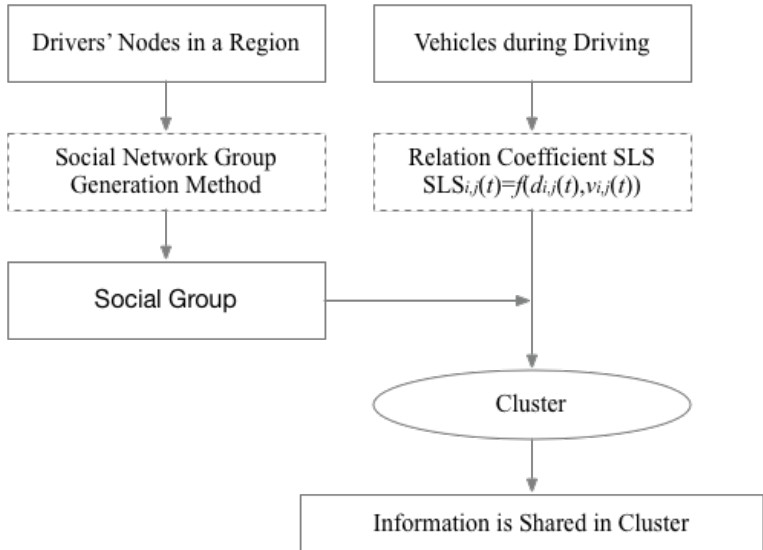

**Figure 3.** The flowchart of the research.

### 2.1. Social Network Group Generation Method

Drivers' group is first generated in the area of the vehicle within the region using a simple social network generation method [17]. The specific steps are stated below.

At every time step, an individual $i$ is selected randomly and updates its status with probability $p_n(t, t_i)$, shown by Equation (1), where $n = n_i$ depending on the individual state, $t$ is the current time, and $t_i$ is the last updated time of $m_i$. The probability that $i$ will not change at the current time is $1 - p_n(t, t_i)$. If $m_i$ has changed, then the individual $i$ is selected and updated in accordance with the following rules:

(1)  If the individual $i$ becomes an isolated state ($m_i=0$), then $i$ will select another isolated individual $j$ with the proportional probability of $p_0(t, t_j)$. Connection between $i$ and $j$ will be established. $m_i$ and $m_j$ will be updated to 1, and $g_i = g_j = \min(i, j)$.

(2)  If the individual $i$ is within a group ($m_i = n > 1$), then $i$ will either leave the group with the probability of $\lambda$ or introduce an isolated individual to the group with the probability of $1 - \lambda$. If $i$ leaves, then $g_i = i$, $m_i = 1$ and all other $i$ within the group will also be updated to their corresponding group and member number. If $i$ chose to introduce a new member $j$ to its group, then, $g_j = g_i$, $m_j \to m_i + 1$ and $m_k \to m_i + 1$, where $k$ is the other number of nodes in the group of $i$:

$$p_n(t, t_i) = \frac{1}{1 + (t - t_i)/N} \tag{1}$$

After forming a social group of drivers, the vehicles' location and velocity information at every moment for clustering are analyzed.

### 2.2. Clustering Method

The relation coefficient of two vehicles' position and speed during each vehicle's moving process is established using enhanced SLS (shown by Equations (2)–(4)) in [12], while SLS is the acronym of Spatial Locality Similarity:

$$SLS_{i,j}(t) = \frac{1}{1 + \sqrt{(\frac{d_{i,j}(t)}{r_{max}})^2 + (\frac{v_{i,j}(t)}{2v_{max}})^2}} \tag{2}$$

$$d_{i,j}(t) = \sqrt{(|x_i(t)| - |x_j(t)|)^2 + (|y_i(t)| - |y_j(t)|)^2} \tag{3}$$

$$v_{i,j}(t) = \sqrt{|v_i(t)|^2 + |v_j(t)|^2 - 2|v_i(t)||v_j(t)|\cos\theta} \tag{4}$$

where $d$ is the distance between two vehicle nodes, $v$ is the speed of the current vehicle, and $\theta$ is the angle of the speed between two nodes. Vehicles are divided into clusters if the SLS is more than $\mu \in [0, 1]$. For a condition in which one vehicle belongs to more than one cluster, this vehicle with different clusters is calculated and chooses a large coefficient cluster to retain. As two vehicles in a same cluster can find each other with one hop, their maximum hop is 2. The equation of the coefficient between node and cluster is expressed as Equation (5):

$$SLS_{i,C}(t) = \frac{n_C}{\sum_{I_r < r} SLS_{i,I} + \frac{1}{2}\sum_{II_r > r} SLS_{i,II}} \tag{5}$$

where $n_C$ is the number of nodes in the current cluster, $SLS_{i,I}$ is the associated degree of the current node with the other nodes in the cluster (the node of the same cluster and within its communication range), and $SLS_{i,II}$ is the associated degree of the current node with other two hops nodes in the cluster. In addition, the other adjacent nodes are introduced into the current cluster if they are in the same group with the current node or the node in the current cluster.

*2.3. Communication Process*

The vehicles communication is narrowed from within the transmission range to within the same cluster to reduce the amount of information transmitted without affecting the transmission effect. The communication process is assumed because the vehicle receives the information transmitted only within the same cluster, not within the transmission range. For different means of communication, the benefit of application layer function $S_i$ is defined; it measures the worth underlying the received broadcast packet from the standpoint of the current hop $i$, and is expressed by Equation (6):

$$S_i(t) = F_i \times \omega_i + R_i \times (1 - \omega_i) \tag{6}$$

$$F_i = e^{-\alpha \times (\hat{D}_i + \gamma)^2} \tag{7}$$

$$R_i = e^{-\beta \times ((\hat{T}_i + T_i) \times v)^2} = e^{-\beta \times (N(i)T \times v(i))^2} \tag{8}$$

$$T = T_H / R_d + E[P] / R_d + DIFS + \delta \tag{9}$$

where $\hat{D}$ is the accumulated distance between the current node and other communication nodes, $\gamma$ is the radius of transmission. $T$ is the delay, $N(i)$ is the number of nodes where the cluster of the current node belongs to, and $v$ is the speed of the current vehicle node. Then, $\omega$, $\alpha$, $\beta$ are influencing factors, while $0 < \omega < 1$ is a weight coefficient that is customized by each relaying participant $i$ according to its favor, $\alpha \geq 0$, is an influence coefficient that is subject to the served application preference and characterizes the desired dissemination distance. If the carried application requires long-range information dissemination, $\alpha$ should be set relatively high. The extreme case of $\alpha = 0$ implies the application does not care about the distance range at all. In addition, $\beta \geq 0$ is the accuracy coefficient that is also subject to the served application preference and reflects the tolerant sensitivity of application to the time-lag effect. If the carried application hopes more accuracy, $\beta$ should be set relatively high. The extreme case of $\beta = 0$ means the distance deviation does not really matter to the application utility at all. $R_d$ is the probability of system to transmit data, E[P] is the packet length, DIFS is the distributed inter-frame space, and $\delta$ is the slot time. Header files contain the physical and MAC layer head messages that are $T_H = PHY_{hdr} + MAC_{hdr}$. The throughput is then calculated as follows [18,19]:

$$S = \begin{cases} \dfrac{np_s \cdot E[p]}{(1 - p_b)t_{slot} + p_b t_{tr}}, & \rho = \dfrac{\lambda}{\mu} > 1 \\ n\lambda \cdot p_s \cdot E[p], & \rho = \dfrac{\lambda}{\mu} \leq 1 \end{cases} \tag{10}$$

$$p_s = p_t(1 - p_0)[1 - p_t(1 - p_0)]^{n-1} \tag{11}$$

$$p_b = 1 - [1 - p_t(1 - p_0)]^n \tag{12}$$

$$p_t = 2/(CW + 1) \tag{13}$$

where $CW$ is the contention window ($cw = 2^i - 1, i = 4, 5, 6, 7, 8, 9, 10$), $p_s$ is the probability of successful transmission of broadcast message, $p_b$ is the probability of the busy channel, $t_{slot}$ is the time slot, $t_{tr}$ is the time of sending a broadcast packet, $p_t$ is the probability of the node sending broadcast, and $p_0$ is the probability of no packet to send, i.e., the probability of the empty packet queue.

*2.4. Simulations*

In the experimental time, it first simulated the driving process of the vehicle by MATLAB, divided the grid system into small regions by the level of the Quadkey method. Then, 1000 vehicles moving processes are performed over a 200 × 200 grid system in VANETMobiSim, and the initial and target positions of the 1000 vehicles are predefined. All the parameters are initialized according to Table 1, and this clustering method can be applied to any vehicle with initial location, vehicle nodes are grouped at every region, and clustered every time step of the moving moment.

**Table 1.** Parameters in simulation.

| Parameter | Meaning | Value |
|-----------|---------|-------|
| DIFS | DCF inter-frame space | 58 μs |
| $\omega$ | Weight coefficient in S | 0.5 |
| $\alpha$ | Influence coefficient | $5 \times 10^{-7}$ |
| $\beta$ | Accuracy coefficient | $5 \times 10^{-7}$ |
| E[P] | Length of the packet data | 1000 × 8 bits |
| $T_H$ | Length of the packet header | 24 × 8 bits |
| $\delta$ | Slot time | 1 μs |
| $R_d$ | System transmission data rate | 6 Mbps |

## 3. Results

Table 1 lists the parameters settings during the simulation. Figures 3 and 4 show the group relationship and cluster of No. 1 vehicle in the region of 01 at the first-time step. Figure 4 shows the social relationship of node 1 from intimacy to alienation by decreasing the color from dark to light, and, in Figure 5, the red node represents the vehicle node 1 and the blue nodes are isolated nodes within the same group, while Figure 5 shows the real-time state of road traffic, and the horizontal and vertical coordinates represent the unit of road length in the simulator.

The clustering method's (expressed as SLS in the result) application interest and communication parameters (e.g., throughput and channel utilization) with those of traditional broadcast communication (expressed as COM in the result) and existing literature (Cluster Management Table, CMT) [11] are compared. Figure 6 depicts the application profit calculated by three communication methods. The application benefit of the new clustering method is higher than the others. In addition, the three curves rise at the same trend after the 2 s. The cluster after this time is mainly determined by the vehicle's location and velocity. Less cluster nodes result in higher benefits. This phenomenon proved that our clustering method can maintain the original information efficiently. The comparison in Figure 7 reveals that the clustering algorithm proposed in this paper has lower channel utilization than the other two algorithms. The results in Figures 8 and 9 indicate that the proposed algorithm has higher throughput in saturated and unsaturated states, which is significantly higher than those of the other two algorithms.

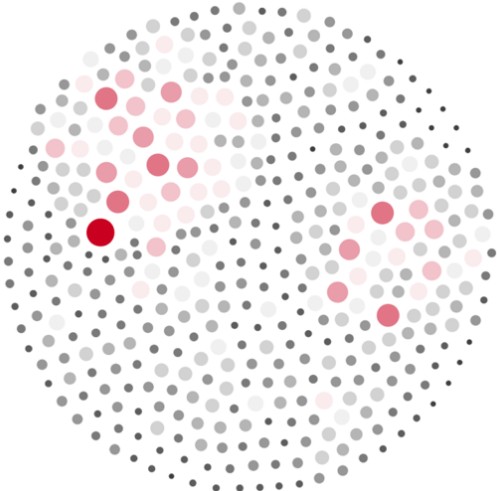

**Figure 4.** A demo of group relationship.

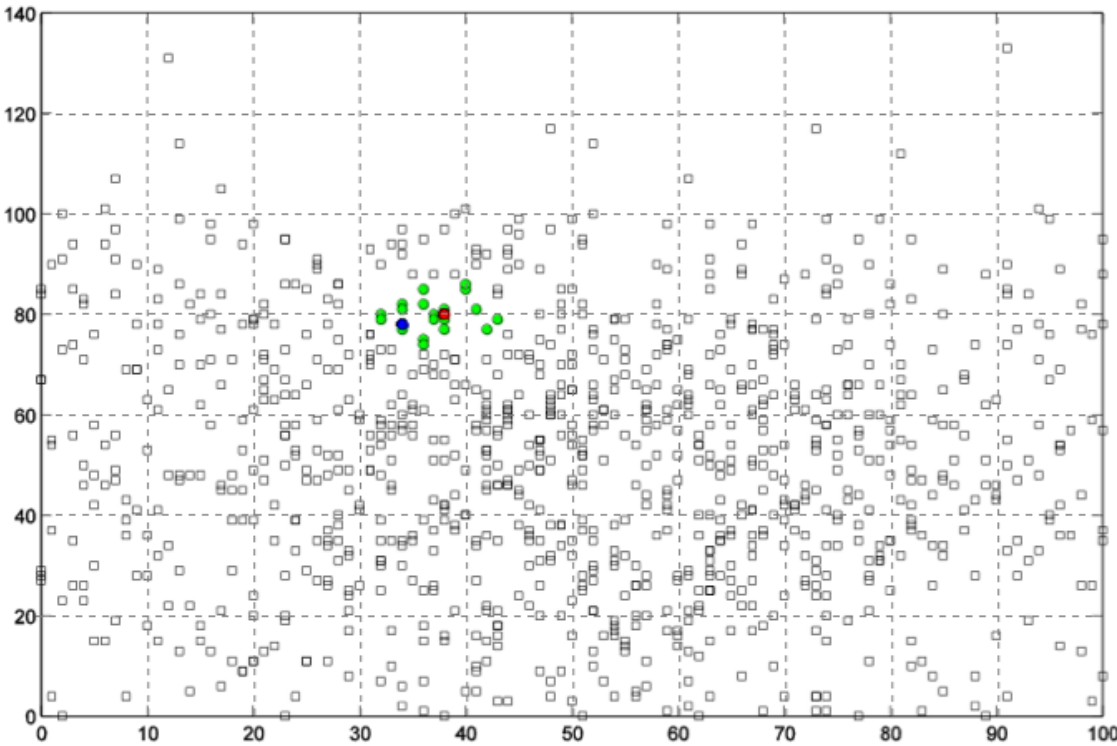

**Figure 5.** A demo of clustering.

The communication parameters under different μ and traditional communication also be compared in the simulation. As shown in Figure 10, the impact of clustering to communication is not significant, and it is basically flat when μ < 0.6, but an evident relationship can be observed when μ > 0.6. Channel utilization decreases as μ increases, as shown in  Figure 11. From Figures 12 and 13, one can know that, when μ is small, the throughput is similar with the traditional communication method, but, when μ is large, it is proportional to the throughput.

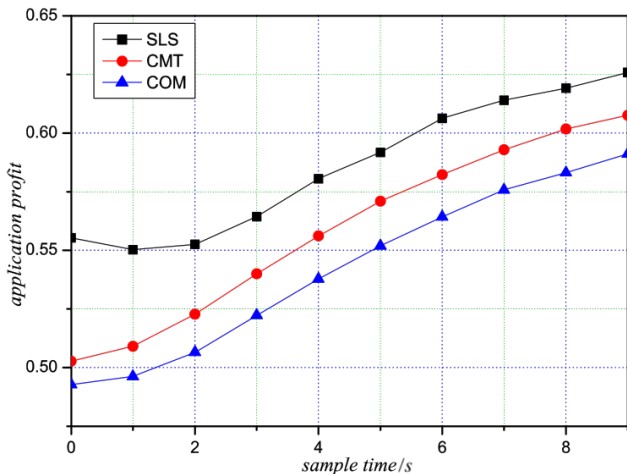

**Figure 6.** Application benefit of the proposed method and other methods.

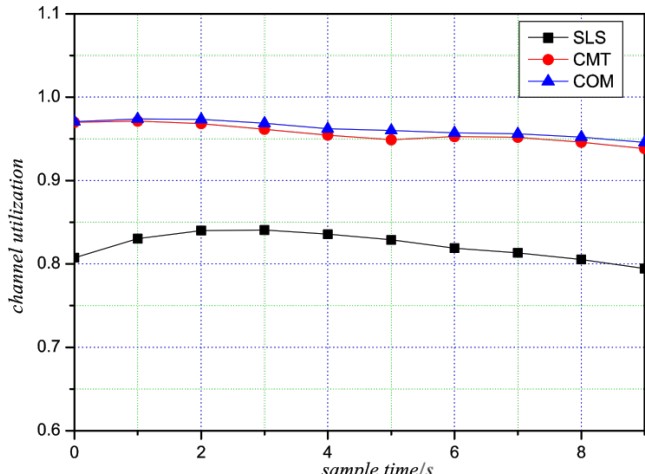

**Figure 7.** Channel utilization of the proposed method and other methods.

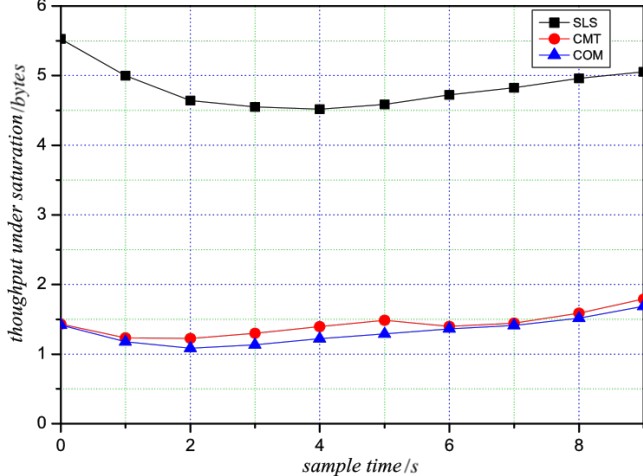

**Figure 8.** Throughput of the proposed method and other methods under saturation.

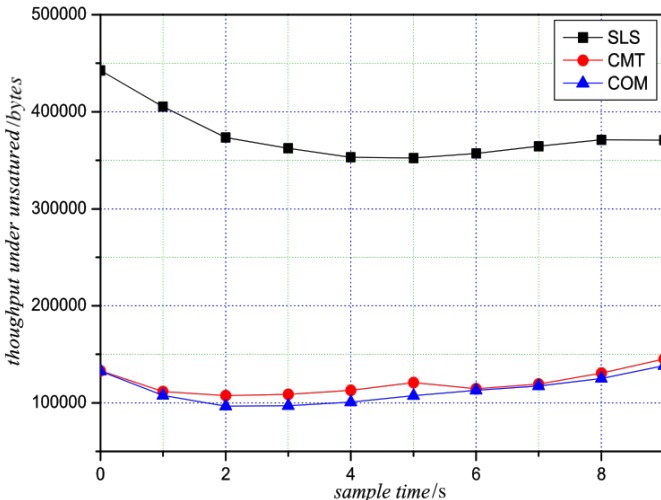

**Figure 9.** Throughput of the proposed method and other methods under unsaturation.

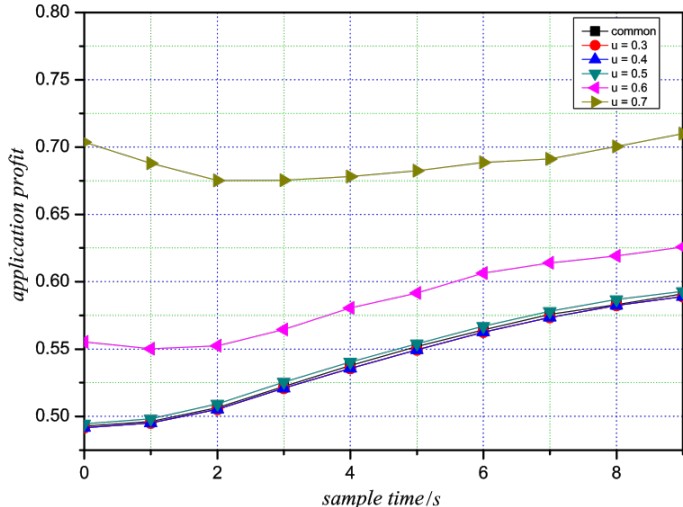

**Figure 10.** Application benefit compared by different values of μ.

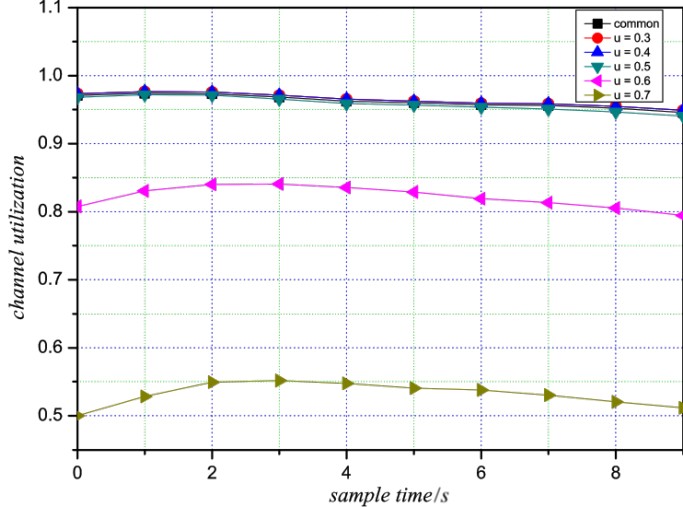

**Figure 11.** Channel utilization compared by different values of μ.

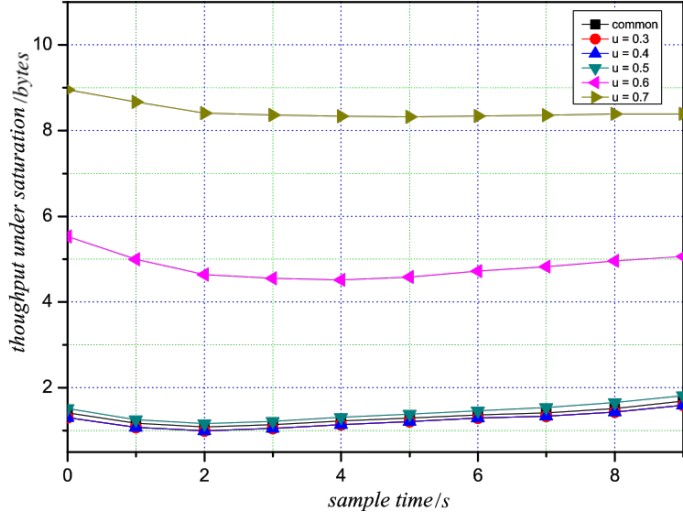

**Figure 12.** Throughput under different values of μ under saturation.

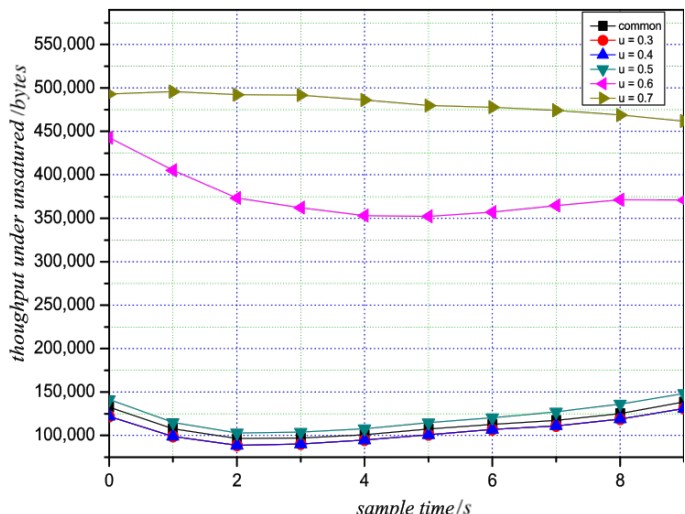

**Figure 13.** Throughput under different values of μ under unsaturation.

## 4. Discussion

In this paper, a new clustering method of nodes in VANET was proposed by incorporating the social relationships of vehicle nodes to assist clustering. There have existed several studies related to the subject of our paper. In [2], the clustering method of MANET is simply transferred and redefined in VANET, in which each node must join in one cluster centered by the nearest base station. Since all the nodes in the cluster of one base station must exchange information with each other via the base station, it is obvious that adding other information (such as social relationships of nodes) in the method is not able to improve the performance of the method. In [1,9–11], the authors proposed a kind of vehicle clustering methods, in which the nodes are clustered by means of the dynamic information (geographic locations trajectories and speeds of nodes). Although the dynamic information has been employed to cluster the nodes in VANET, the social relationships between nodes derived by drivers fail to be considered in these methods. To the best of our knowledge, however, many studies have demonstrated that human dynamics is a very significant factor in the social network of the people involved. As a network involved drivers or users, VANET thus should be investigated by means of social relationships of drivers or users. In the method proposed in this paper, the social relationships of nodes are used to modify the clusters results obtained by locations and speeds, which makes it reasonable that the proposed method is superior to the one using the neighborhood of base station in [2] and the ones clustering vehicle nodes by dynamic information in [1,9–11].

After constructing the clusters of all nodes, each node transmits information to other nodes within its cluster. Thus, the method of clustering nodes significantly affects the profit of communication in VANET. Compared with the broadcast method (transmitting information within the range of broadcast) and the representative method [11] (transmitting information within the cluster generated by relative speed), our method can reach more throughput and higher communication efficiency than them from the perspective of simulation. The fundamental reason is that social relationships are combined into the process of clustering nodes in VANET.

However, by our method, only a few nodes are added in the cluster of one node by using the social relationships of the node. It should be because the social relationships are generated by random connection of nodes, and are incorporated into the process of clustering of nodes by modifying the cluster of one node based on its group of social relationships. Therefore, how to characterize the social relationships of nodes and how to utilize the social relationship for clustering nodes will be worth doing further research in the field of VANET.

## 5. Conclusions

This paper proposed a new clustering technique based on social group by introducing the social dynamic of drivers into vehicle network clustering to improve vehicle communication. The method combined vehicle nodes' social group with position and speed, resulting in a clustering method in the communication process. As can be seen from Section 3, in the process of vehicle simulation, the vehicle cluster is no longer a simple adjacent vehicle nodes but also nodes with a social relationship. By defining and simulating the vehicle communication process, it concluded that this proposed clustering method can enhance the interests of the application layer and throughput of vehicles communication. However, as the experimental results shown, in the process of vehicle driving, the probability of social nodes appearing in the cluster is not very high, which also conforms to the law of vehicle driving. The next research work will tackle other ways of forming social relations and clustering based on the different drivers' community and real-time location and speed.

**Author Contributions:** L.L. contributed the idea, designed the system model, checked the results, and wrote the manuscript. W.W. prepared the manuscript. Z.G. was responsible for formulating the research issues and revising the paper. All authors have read and agreed to the published version of the manuscript.

**Funding:** This work was supported by the Natural Science Foundation of Shanxi Province, China (201801D221165), Scientific and Technological Innovation Programs of Higher Education Institutions in Shanxi, China (2019L0057), the National Nature Science Foundation (61673249, U1805263), the Key R&D program of Shanxi Province (International Cooperation No.201903D421050).

**Conflicts of Interest:** The authors declare no conflict of interest.

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
