# Peer review of "Driver’s Social Relationship Based Clustering and Transmission in Vehicle Ad Hoc Networks (VANETs)"

_electronics, doi:10.3390/electronics9020298_

Round 1

Reviewer 1 Report

The paper got better after the reviews and clarifications presented. Related to the Figures, please clarify and complete the legends. For example: Figure 9, throughput under unsaturated as function of sample time, what is the association of the throughput to a metric unit? What does a value of 400000 mean for example? It is not clear to those who read the article. The same problem occurs in all figures.

Reviewer 2 Report

I have reviewed the Manuscript ID: electronics-716913, representing the revised version of the Manuscript ID: electronics-568277, having the title "Driver’s Social Relationship Based Clustering and Transmission in Vehicle Ad-hoc Networks (VANETs)" and I can conclude that even if overall the manuscript has been improved, the authors did not address all the signaled issues and have actually procrastinated, postponed some aspects. Therefore, I have devised and wrote the following comments to the authors of the manuscript under review:

The "Discussion" section. The current form of the "Discussion" section is not appropriate as it contains only 19 lines and it does not mention a series of important elements required by this section. For example, in order to validate the usefulness of their research, in the "Discussion" section, the authors should make a comparison between their approach from the manuscript and other similar ones that have been developed in the literature for the same or related purposes. In the actual form of the manuscript, the "Discussion" section contains only one reference to other studies, so the comparison is insufficiently approached in the manuscript's current form. There are many other studies in the scientific literature related to the subject of the manuscript to which the authors can compare to and this comparison will highlight even more the novel aspects that their paper brought in contrast to the existing studies. This comparison is mandatory in order to highlight the clear contribution to the current state of knowledge that the authors brought. The authors did not address this issue in the revised version of the manuscript even if I have signaled it in my previous review reports.

The "Discussion" section. I consider that the paper will benefit if the authors make a step further, beyond their analysis and provide an insight at the end of the "Discussion" section regarding what they consider to be, based on the obtained results, the most important, appropriate and concrete actions that the decisional factors and all the involved parties should take in order to benefit from the results of the research conducted within the manuscript. The authors did not address this issue in the revised version of the manuscript even if I have signaled it in my previous review reports.

Other remarks:

Lines 94-95: "…N (1,2,3,. . .N) is the number of vehicles within their region; G (1,2,3,. . .G) is the initial group number of the node…" The authors must explain in more details what is the meaning of the two pairs of parentheses within these lines.

Lines 184-186: "In the figures below, Application profit (defined by the equation 6, shown in Figure 6 and Figure 10)” and “Channel utilization (defined by the equation 12, shown in Figure 7 and Figure 11)” are proportion, and there are no ordinate units. Then, the “throughput (defined by the equation 10, shown in Figure 8,9,12,13)” is measured by byte.".  What is the meaning of the quotation marks used within these lines?

Lines 119-121: "SLS(shows by equation 2-4) in [12], while SLS is the acronym of Spatial Locality Similarity, which quote from the literature of “Internode mobility correlation for group detection and analysis in VANETs." What is the meaning of the quotation marks used within these lines and why they are not in pair?

At Figures 6-13, the measurement units of the axes are missing.

Round 2

Reviewer 2 Report

I have reviewed the Manuscript ID: electronics-716913, having the title "Driver’s Social Relationship Based Clustering and Transmission in Vehicle Ad-hoc Networks (VANETs)" and I can conclude that the authors have addressed the signaled issues, therefore improving the manuscript.

This manuscript is a resubmission of an earlier submission. The following is a list of the peer review reports and author responses from that submission.

Round 1

Reviewer 1 Report

The article describes a new method for clustering vehicle networks. I find the main idea interesting and worth investigating, but the paper must be significantly improved before publication.

My main concern is about quality of presentation, because it prevents the reader from understanding what was done by the authors, how it was done, and what's the significance of it. More detailed comments are following.

Major:

1) Throughout the article, extensive editing of English language and style must be performed.

2) The literature review should be revised, extended and improved. After reading it, I would like to have an idea about what are the main direction of research in the field, what is already done, what is lacking, and how authors' work helps to address it. In particular, it should be better explained what is the relation between MANETs and VANETs, and how and if methods developed for one could be applied to the other. I also wonder, if there is any recent research on the topic, as the most recent works cited are from 2016 and 2015.

3) The explanation of the method must be improved. There are many small ... (see minor comments) that together make the method hard to understand. Authors should explain where these equations come from.

4) Figures and their descriptions should be improved (see minor comments).

Minor:

1) For an unfamiliar reader authors should specify what is "mobile network" and how it differs from vehicle network.

2) Second paragraph of introduction refers to many different studies but doesn't provide any citation. All references should be cited accordingly.

3) What is "the interests of the application layer"?

4) What does VANET stands for?

5) Authors should provide better description of what is Quadkey. Or at least reference Figure 2 in text. Also, it’s not clear, if level 2 corresponds to level 2 in the figure. From figure 2 it seems like level 2 encompasses the whole east-south-east Asia.

6) When describing clustering method, what is the difference between vehicle's group and cluster?

7) What is SLS?

8) Authors should clearly specify what are the free parameters of their clustering model.

9) Relation and correlation mean different things and should not be used interchangeably.

10) All variables used in formulas should be explained.

11) Figures 3 and 4 should have caption that would allow to understand them. What does color, size and shade of the circle mean? What are the axes in figure 4? Figure 4 is too small. I don't see any blue nodes in Figure 3.

12) What do SLS, CMT and COM labels mean?

13) What are the units of y-axis in figures 5-12?

Author Response

Thank you very much for your meticulous inspection on the paper. The revision was carried out as a result of the comments of three reviewers. In our point-by-point response attached below, reviewer's  comments are in Black color, and our responses are in Red color. We are looking forward to hearing from you. Thank you.

Reviewer 2 Report

Avoid constantly using the term "we", write in the 3rd person.

The paper needs to be more clear, simulation platform and constraints such all specifications of scenarios needs to be described. Conclusion needs to be improved.

Author Response

Thank you very much for your meticulous inspection on the paper. The revision was carried out as a result of the comments of three reviewers. In our point-by-point response attached below, reviewer’s comments are in Black color, and our responses are in Red color. We are looking forward to hearing from you. Thank you.

Reviewer 3 Report

Dear Esteemed Editors,

I have reviewed the Manuscript ID: electronics-568277, with the title "Driver's Social Relationship Based Clustering and Transmission in VANET". In this paper, the authors analyze a clustering method for vehicle networks that is based on drivers' social relationship combined with the instantaneous position and speed of the vehicle node. The authors state that the simulation results showed that the proposed clustering method can improve the effectiveness of information transmission and increase the utilization of the application layer. I consider that the paper will benefit if the authors address within the manuscript the following aspects:

Overall comments regarding the manuscript: The Main Strong Point: The manuscript under review approaches a very interesting topic for the experts in the field. The Main Weak Point: I consider that the main weak point consists in the structure of the manuscript. If the authors make an effort to improve the structure of the manuscript and complement it with the recommendations from the specific comments, the authors will arrive at an article that can bring a valuable insight to the current state of knowledge. The sections of the manuscript. In the actual form of the paper, its sections are not according to the ones recommended by the Electronics MDPI Journal's Template. The manuscript under review will benefit if it is restructured in accordance with the above-mentioned template that provides a more logical structure that is much more appropriate for a research article. The restructuring of the manuscript will also help the authors to express better the novelty of their work and the contribution that they have made to the current state of knowledge. Consequently, the manuscript under review should be restructured as follows: Abstract, Keywords, 1. Introduction, 2. Materials and Methods, 3. Results, 4. Discussion, 5. Conclusions (not mandatory), 6. Patents (not mandatory), Supplementary Materials (not mandatory), Author Contributions, Funding, Acknowledgments, Conflicts of Interest, Appendices and References. The "Abstract" of the paper. Along with the elements already presented, in the abstract the authors should also declare and briefly justify the novelty of their work. The "Introduction" and the "Related work" sections. The "Related work" section's purpose and the one of the "Introduction" are overlapping and therefore the two sections should be concatenated and reorganized. In the current form of the manuscript, these two sections contain a series of cited papers. I do not contradict the value of these papers, or their relevance in this context, but I consider that the article under review will benefit if the authors extend these sections by analyzing appropriately the cited papers and by analyzing more other papers in order to contextualize their study. The literature review of the cited papers has been performed in the following manner: "Fig.1 shows an example of clusters in the traditional mobile network clustering[2], In [3-5], different cluster head selection algorithms are studied and how to choose cluster head in a particular cluster is analyzed to extend network lifetime and improve the reliability of data transmission. Bednarczyk et al. [6] proposed a clustering method based on weight coefficients in MANETs. Rao et al. [7] devised a method based on scalable and energy efficient multipath routing protocol for MANETs. Rajkumar et al. [8] suggested a clustering based on geographical factors in MANETs." (Lines 52-57). I consider that it is not appropriate for the manuscript to cover 7 (like the authors have did) or even more scientific works in a few lines just for the sake of obtaining an appropriate size of the References section. In the "Introduction" section, the authors must introduce a presentation of the current state of the research field by reviewing it carefully and by citing other key publications. By doing so, the problem will be put into context and it will benefit the readers as well. The purpose of the literature survey is to highlight for the involved referenced papers the main contribution that the authors of the referenced papers have brought to the current state of knowledge, the methods used by the authors of the referenced papers, a brief presentation of the main obtained results and some limitations of the referenced article. This is the only way to contextualize the current state of the art in which the authors of the manuscript position their paper and address aspects that have not been tackled/solved yet by the existing studies from the body of knowledge in contrast with the manuscript under review. The "Materials and Methods" section. In the actual form of the manuscript, the "Materials and Methods" section is missing. It will benefit the paper if the authors restructure their paper and devise a proper "Materials and Methods" section, as requested by the Electronics MDPI Journal's Template. I consider that the authors must pay more attention to the appropriate citation of the methods and results that have been retrieved from the scientific literature. When the authors present the information in the "Materials and Methods" section, they must assume clearly their original contribution by specifying this fact and by highlighting the fact that starting from a certain point there are depicted the original and novel aspects of their research. The "Materials and Methods" section. In order to help the readers better understand the methodology of the conducted study, in the "Materials and Methods" section, the authors should devise a flowchart that depicts the steps that they have processed in developing their research and most important of all, the final target. This flowchart will facilitate the understanding of the proposed approach and it will make the article more interesting to the reader if used as a graphical abstract. The "Materials and Methods" section. In addition to the flowchart, in order to bring a benefit to the manuscript, the authors should state and justify very clear in the "Materials and Methods" section the choices they have made when developing the final form of their proposed approach. The authors should state what has justified using this approach, what is special, unexpected, or different in their research methodology. It will benefit if the authors mention if they have tried other approaches that in the end led them to the current form of their research design. The "Materials and Methods" section. I consider that in the "Materials and Methods" section the authors should specify the detailed hardware and software configurations that they have used when developing their research, in order to provide all the necessary details for assuring the reproducibility of their study. The equations. All the equations within the manuscript should be explained, demonstrated or cited, as there are some equations that have not been introduced in the literature for the first time by the authors and that are not cited. The "Discussion" section. In order to validate the usefulness of their research, in the "Discussion" section (which is currently missing from the manuscript), the authors should make a comparison between their approach from the manuscript and other similar ones that have been developed in the literature for the same or related purposes. There are a lot of valuable studies in the scientific literature related to the subject of the manuscript to which the authors can compare to and this comparison will highlight even more the novel aspects that their paper brought in contrast to the existing studies. The "Discussion" section. The authors should present the findings and their main implications in the "Discussion" section, also highlighting current limitations of their study, and briefly mention some precise directions that they intend to follow in their future research work. The "Discussion" section. I consider that the paper will benefit if the authors make a step further, beyond their analysis and provide an insight at the end of the "Discussion" section regarding what they consider to be, based on the obtained results, the most important, appropriate and concrete actions that the decisional factors and all the involved parties should take in order to benefit from the results of the research conducted within the manuscript. Lines 134-135, the "Simulation and Result" section. " We first simulated the driving process of the vehicle. We simulated 1000 vehicle moving processes in a 200x200 grid system." Can the authors mention to which extent their clustering method can be easily applied to other situations that impose different parameters? In this way, the authors could highlight more the generalization capability of their approach in order to be able to justify a wider contribution that has been brought to the current state of art. Other remarks. Lines 2-4, the title of the manuscript: "Driver's Social Relationship Based Clustering and Transmission in VANET". Acronyms must be avoided in the title, even if they are widely known. The title could be modified under the form "Driver’s Social Relationship Based Clustering and Transmission in Vehicular Ad-hoc Networks (VANETs)". Regarding the other acronyms used in the manuscript, they should be explained the first time when they are introduced. The Figures within the manuscript. According to the Electronics MDPI Journal's Template, all the figures should be cited in the main text as "Figure 1", "Figure 2", etc. In the manuscript under review, in the main text the figures' citations appear under the form "Fig. 1", "Fig. 2". Please address this issue by modifying the way in which the figures are referred in the main text and their captions, according to the Electronics MDPI Journal's Template. Figures 2 and 10 are not referred in the text of the manuscript.

Author Response

(The authors gave the same response as above.)

Round 2

Reviewer 1 Report

Most of my previous comments were not addressed. Quality of presentation remained poor with some newly added mistakes. Literature review remained brief and incomplete. (This point was well explained by another reviewer and was completely ignored by the authors, as well as many other points.) The method is still not understandable by an unfamiliar reader.

Changing the narrative to 3rd person didn't go smoothly and added inconsistencies.

The part about article's structure didn't change, although the structure did change.

Newly added conclusion "This study demonstrates that social relations have an impact on communication and driving efficiency during the vehicle driving." is not supported by results. There is nothing in paper about driving efficiency.

Lines 49-50. Authors claim that something is CURRENTLY being studied and cite a research from 2014.

In their response, authors say: "Response 7: We appended a detail explanations of SLS....". I don't see this in the paper.

"... correlation between position and speed ..." is not mentioned in method, so I wonder, how is it used?

"...in Figure 4, the red node represents the vehicle node 1 and the blue nodes are isolated nodes within the same group." - I don't see any red or blue nodes in Figure 4.

"Figure 4 shows the real-time state of road traffic, and the horizontal and vertical coordinates represent the unit of road length in the simulator." - This should be mentioned in text. Also, figure must be enlarged, axis should be labeled.

“The labels of SLS, CMT and COM mean the clustering method of we proposed, the method of existing literature of [3] (numbered by [11] in the revision), and traditional broadcast communication respectively”. - I don't see this in the main text.

“Application interest (defined by the equation 6, shown in Figure 5, 9)” and “Channel utilization (defined by the equation 12, shown in Figure 6, 10)” are proportion, and there are no ordinate units. Then, the “throughput (defined by the equation 10, shown in Figure 7,8,11,12)” is measured by byte. - This should be mentioned in text and not just in the response.

Reviewer 2 Report

The paper has been improved and some points have been clarified. However, the scientific innovations of work remain unclear.
Instead of a conclusion, you only present one section for discussion, however, a conclusion is important to understand the scientific contributions of the paper and to properly finish the document.

Reviewer 3 Report

I have reviewed the revised version of the Manuscript ID: electronics-568277, having the first title "Driver's Social Relationship Based Clustering and Transmission in VANET" and the revised title "Driver’s Social Relationship Based Clustering and Transmission in Vehicle Ad-hoc Networks (VANETs)" and I can conclude that even if overall the manuscript has been improved, the authors did not address all the signaled issues and have actually procrastinated, postponed some aspects.  Therefore, I have devised and wrote the following comments to the authors of the manuscript under review:

The "Abstract" of the paper. Along with the elements already presented, in the abstract the authors should also declare and briefly justify the novelty of their work. The authors did not address this issue in the revised version of the manuscript even if I have signaled it in my previous review report.

The "Introduction" section. In the current form of the manuscript, this section contains a series of cited papers. I do not contradict the value of these papers, or their relevance in this context, but I consider that the article under review will benefit if the authors extend these sections by analyzing appropriately the cited papers and by analyzing more other papers in order to contextualize their study. The literature review of the cited papers has been performed in the following manner: "Figure 1 shows an example of clusters in the traditional mobile network clustering [5], In [6-8], different cluster head selection algorithms are studied and how to choose cluster head in a particular cluster is analyzed to extend network lifetime and  improve the reliability of data transmission. Bednarczyk et al. [9] proposed a clustering method based on weight coefficients in MANETs. Rajkumar et al. [10] suggested a clustering based on geographical factors in MANETs." (Lines 42-47). I consider that it is not appropriate for the manuscript to cover 6 (like the authors have did) or even more scientific works in a few lines just for the sake of obtaining an appropriate size of the References section. In the "Introduction" section, the authors must introduce a presentation of the current state of the research field by reviewing it carefully and by citing other key publications. By doing so, the problem will be put into context and it will benefit the readers as well. The purpose of the literature survey is to highlight for the involved referenced papers the main contribution that the authors of the referenced papers have brought to the current state of knowledge, the methods used by the authors of the referenced papers, a brief presentation of the main obtained results and some limitations of the referenced article. This is the only way to contextualize the current state of the art in which the authors of the manuscript position their paper and address aspects that have not been tackled/solved yet by the existing studies from the body of knowledge in contrast with the manuscript under review. The authors did not address this issue in the revised version of the manuscript even if I have signaled it in my previous review report.

The "Materials and Methods" section. In order to help the readers better understand the methodology of the conducted study, in the "Materials and Methods" section, the authors should devise a flowchart that depicts the steps that they have processed in developing their research and most important of all, the final target. This flowchart will facilitate the understanding of the proposed approach and it will make the article more interesting to the reader if used as a graphical abstract. The authors did not address this issue in the revised version of the manuscript even if I have signaled it in my previous review report.

The "Materials and Methods" section. I consider that in the "Materials and Methods" section the authors should specify the detailed hardware and software configurations that they have used when developing their research, in order to provide all the necessary details for assuring the reproducibility of their study. The authors did not address this issue in the revised version of the manuscript even if I have signaled it in my previous review report.

The equations. All the equations within the manuscript should be explained, demonstrated or cited, as there are some equations that have not been introduced in the literature for the first time by the authors and that are not cited. The authors did not address this issue in the revised version of the manuscript even if I have signaled it in my previous review report.

 The "Discussion" section. The current form of the "Discussion" section is not appropriate as it contains only 8 lines and it does not mention a series of important elements required by this section. For example, in order to validate the usefulness of their research, in the "Discussion" section, the authors should make a comparison between their approach from the manuscript and other similar ones that have been developed in the literature for the same or related purposes. There are a lot of valuable studies in the scientific literature related to the subject of the manuscript to which the authors can compare to and this comparison will highlight even more the novel aspects that their paper brought in contrast to the existing studies. The authors did not address this issue in the revised version of the manuscript even if I have signaled it in my previous review report.

The "Discussion" section. The authors should present the findings and their main implications in the "Discussion" section, also highlighting current limitations of their study, and briefly mention some precise directions that they intend to follow in their future research work. The authors did not address this issue in the revised version of the manuscript even if I have signaled it in my previous review report.

The "Discussion" section. I consider that the paper will benefit if the authors make a step further, beyond their analysis and provide an insight at the end of the "Discussion" section regarding what they consider to be, based on the obtained results, the most important, appropriate and concrete actions that the decisional factors and all the involved parties should take in order to benefit from the results of the research conducted within the manuscript. The authors did not address this issue in the revised version of the manuscript even if I have signaled it in my previous review report.